# Structural and Biochemical Analysis of the Dual Inhibition of MG-132 against SARS-CoV-2 Main Protease (Mpro/3CLpro) and Human Cathepsin-L

**DOI:** 10.3390/ijms222111779

**Published:** 2021-10-29

**Authors:** Elisa Costanzi, Maria Kuzikov, Francesca Esposito, Simone Albani, Nicola Demitri, Barbara Giabbai, Marianna Camasta, Enzo Tramontano, Giulia Rossetti, Andrea Zaliani, Paola Storici

**Affiliations:** 1Elettra—Sincrotrone Trieste, 34149 Trieste, Italy; elisa.costanzi@elettra.eu (E.C.); nicola.demitri@elettra.eu (N.D.); barbara.giabbai@gmail.com (B.G.); 2Fraunhofer Institute for Translational Medicine and Pharmacology (ITMP), 22525 Hamburg, Germany; Maria.Kuzikov@itmp.fraunhofer.de (M.K.); Andrea.Zaliani@itmp.fraunhofer.de (A.Z.); 3Department of Life Sciences and Chemistry, Jacobs University Bremen GmbH, 28759 Bremen, Germany; 4Department of Life and Environmental Sciences, University of Cagliari, 09124 Cagliari, Italy; francescaesposito@unica.it (F.E.); marianna.camasta1997@gmail.com (M.C.); tramon@unica.it (E.T.); 5Institute for Neuroscience and Medicine (INM-9) and Institute for Advanced Simulations (IAS-5) “Computational Biomedicine”, Forschungszentrum Jülich, 52425 Jülich, Germany; s.albani@fz-juelich.de (S.A.); g.rossetti@fz-juelich.de (G.R.); 6Department of Biology, Faculty of Mathematics, Computer Science and Natural Sciences, RWTH Aachen University, 52062 Aachen, Germany; 7Jülich Supercomputing Centre (JSC), Forschungszentrum Jülich, 52425 Jülich, Germany; 8Department of Neurology, Faculty of Medicine, RWTH Aachen University, 52074 Aachen, Germany

**Keywords:** SARS-CoV-2, Mpro/3CLPro, Cathepsin-L, peptidomimetics, MG-132, dual target inhibitor

## Abstract

After almost two years from its first evidence, the COVID-19 pandemic continues to afflict people worldwide, highlighting the need for multiple antiviral strategies. SARS-CoV-2 main protease (Mpro/3CLpro) is a recognized promising target for the development of effective drugs. Because single target inhibition might not be sufficient to block SARS-CoV-2 infection and replication, multi enzymatic-based therapies may provide a better strategy. Here we present a structural and biochemical characterization of the binding mode of MG-132 to both the main protease of SARS-CoV-2, and to the human Cathepsin-L, suggesting thus an interesting scaffold for the development of double-inhibitors. X-ray diffraction data show that MG-132 well fits into the Mpro active site, forming a covalent bond with Cys145 independently from reducing agents and crystallization conditions. Docking of MG-132 into Cathepsin-L well-matches with a covalent binding to the catalytic cysteine. Accordingly, MG-132 inhibits Cathepsin-L with nanomolar potency and reversibly inhibits Mpro with micromolar potency, but with a prolonged residency time. We compared the apo and MG-132-inhibited structures of Mpro solved in different space groups and we identified a new apo structure that features several similarities with the inhibited ones, offering interesting perspectives for future drug design and in silico efforts.

## 1. Introduction

After more than one and a half years since its first isolation (December 2019) [1] coronavirus SARS-Cov-2 is still threatening world health and has dramatically hampered the lifestyle on a global level. Vaccination campaigns have started but the need for finding an effective drug is still very urgent. To date 241,411,380 confirmed cases of COVID-19, including 4,912,112 confirmed deaths were reported by the WHO [2] (20 October 2021). Although 6,545,309,084 vaccine doses were administered worldwide, the vast majority of countries remain with strict restrictions on daily life, including closures of school, workplace and the culture sector, as well as travel limitations. In addition, the constant appearance of novel variants spreading worldwide urgently highlights the need for multiple antiviral strategies.

The large RNA-genome of SARS-CoV-2 codes for about 16 non-structural proteins, including the 3C-Like or main protease (Mpro; nsp5), the papain-like protease (PLpro; nsp3), the RNA-dependent RNA polymerase (RdRp; nsp12), the helicase (Hel; nsp13), two methyltransferases (guanine-N7-methyltransferase with exoribonuclease activity; nsp14 and nucleoside-2’-O-methyltransferase; nsp16) and four structural proteins (spike (S), envelope (E), membrane (M) and nucleocapsid (N) protein) [3]. Each of these proteins represents a target for the development of antiviral drugs. With exception of the spike protein, the majority of anti-SARS-CoV-2 compounds that are designed up till now are prone to interfere with the viral replication machinery. A significant number of compounds identified and developed so far are directed against the main protease, which is a very attractive target against SARS-CoV-2 [4]. The first Mpro inhibitor (PF-7304814) has entered clinical trials in March 2021 [5,6] together with other protease inhibitors like the DPP1 inhibitor Brensocatib [7] or like the urokinase inhibitor Upamostat [8].

Mpro is responsible for viral maturation by catalyzing the proteolytic processing of the precursor polyproteins pp1a and pp1ab at 11 unique sites. It is highly conserved among different CoVs [9], and as far as it is known, it has no human homolog, and shows a limited mutation rate, making it highly suitable for the development of multi-viral drugs that can interfere with vital cycles of different coronaviruses, including different SARS-CoV-2 variants.

Mpro is a cysteine protease of 33.8 kDa, catalytically active as a stable homodimer, with the two protomers associated at right angles to form a heart-shaped complex. In each protomer three domains are distinguished: the N-terminal domain I (residues 10–99), and the domain II (100–182), which fold together in six antiparallel β-barrels, and the C-terminal domain III (residues 198–303) which is a globular cluster of five helices, mainly contributing to protein dimerization. At the interface between domain I and II, the active site that contains a non-canonical Cys-His dyad is located and is composed of four binding subsites (S1’, S1, S2 and S4) which accommodate substrates from the N terminus to the C terminus with a distinctive cleavage preference for glutamine at the *P* 1 site (Leu-Gln/Ser,Ala,Gly).

As part of the Exscalate4CoV program (“EXaSCalesmArtpLatform Against paThogEns for Corona-Virus, Exscalate4CoV or E4C”, http://www.exscalate4cov.eu, accessed on 28 October 2021), which is funded through EU’s H2020-SC1-PHE-CORONA- VIRUS-2020 emergency call (Grant 101003551), we have reported the results of the 8.7 K compound in-vitro repurposing screen against the SARS-CoV-2 main protease (Mpro) [10]. Among the reported hits, MG-132 showed an IC_50_ = 7.4 μM in the Mpro enzymatic assay, and was one of the few compounds with observed antiviral activity detected with qPCR as reduction of viral RNA (EC50 = 0.1 μM) using Vero-E6 cells infected by the SARS-CoV-2 strain BetaCov/Belgium/GHB-03021/2020 [10]. Of note, we observed that in the enzymatic assay, Mpro was not sensitive to the presence of reducing agents (IC_50_ = 4.7 μM with 1 mM DTT), differently from a number of other inhibitors [10]. As a first hint of the binding mode, we have also reported a docking in the main catalytic cavity of Mpro, suggesting a pose typical for a peptidomimetic.

MG-132 belongs to the class of synthetic peptide aldehydes composed of a tri-Leucine peptide with the N-terminal protected by a benzyloxycarbonyl derivative and the C-terminal carboxylate reduced to an aldehyde. Originally identified as a potent, cell-permeable inhibitor of the chymotryptic activity of the proteasome; it was initially considered as an antineoplastic drug [11,12]. Recently it has been repurposed as an antiviral agent against SARS-CoV-2 [13]. In fact, MG-132 was reported to strongly inhibit the replication of SARS-CoV, the closest relative of SARS-CoV-2 [14]. The effect on viral infection was mainly linked with the inhibition of the host protease, calpain-m [14], and not due to proteasome or autophagy impairment. MG-132 antiviral activity was also observed for other viruses, such as Herpes simplex virus 1 [15], hepatitis E [16], human cytomegalovirus [17], porcine circovirus type 2 [18], and coxsackievirus B3 [19]. In all these cases the control of viral cell-entry was directly related to the interference of the ubiquitin-proteasome system rather than direct inhibition of protease-driven viral replication [15,16,17,18]. A different mechanism was observed for the Hendra virus, a representative of the paramyxoviruses, where inhibition of infection was dependent on proteolytic processing of paramyxovirus fusion proteins that are essential for virus entry. The authors show that Cathepsin-L activity is required for viral entry and MG-132 is able to inhibit this step [20].

Cathepsin-L belongs to a subclass of lysosomal cysteine proteases and shows, among its main functions, the proteolysis of antigens generated by pathogen endocytosis [21]. Recently, a crucial role of Cathepsin-L in COVID-19 patients has been identified based on evidence that its circulating levels are elevated after SARS-CoV-2 infections, and positively correlate with disease severity [22]. Functionally, Cathepsin-L cleaves the furin primed SARS-CoV-2 S protein into smaller fragments and promotes S-protein-mediated cell–cell fusion, efficiently enhancing SARS-CoV-2 infection [22,23,24]. Showing a mechanism similar, but not identical to that previously reported for S protein from SARS-CoV [25].

Several observations are emerging on the importance to impair the viral entry and replication by blocking either the two viral proteases (Mpro and PLpro) and/or the host proteases that drive for activation and penetration of SARS-CoV-2, such as TMPRSS2, furin, Cathepsins-L and Cathepsin-B [26,27].

Considering these data, and the potency observed for MG-132 on viral-infected cells here we provide a detailed structural analysis of the binding mode of MG-132 on SARS-CoV-2 Mpro and a comprehensive modeling analysis for binding motifs in Cathepsin-L, offering structural hints for the future design of an optimized Mpro selective, or more potent, dual inhibitor.

## 2. Results

### 2.1. Structural Characterization of Mpro in Apo and MG-132 Bound Forms

Mpro is relatively easy to produce and prone to crystallize in different conditions as shown by the more than 380 crystal structures deposited in the Protein Data Bank (https://www.wwpdb.org/, accessed on 28 October 2021). In the context of the Exscalate4CoV project, we obtained and tested more than 2000 crystals of Mpro. In the beginning, we reproduced the conditions published by Zhang et al. using the PACT screening [28] and we solved structures of Mpro in the apo form at 1.65 Å resolution in the well-described spacegroups *C* 2 (PDB ID:7ALH) and *P* 2_1_ (PDB ID:7ALI). Crystals in these two spacegroups can be found within equivalent crystallization conditions. The main protease always presents as a dimer with the 2-fold axis being either crystallographic (spacegroup *C* 2) or non-crystallographic (spacegroup *P* 2_1_). In the effort of obtaining structures of Mpro in complex with different inhibitors, we subsequently obtained crystals by seeding techniques that grow in spacegroup *P* 2_1_2_1_2_1_ and have a dimer in the asymmetric unit. First structure of this series, obtained by our group, is that of Mpro in complex with myricetin [10,29]. Subsequently, we solved the apo structure in this same spacegroup (PDB ID: 7BB2). In all three space groups, the apo structures show well-defined electron density throughout the entire polypeptide chains, with exception of the dimer in spacegroup *P* 2_1_2_1_2_1_ where only the chain A could be well traced till the last residue Gln306, while the chain B could be modeled till Ser301. Even though the general heart-shaped fold of the dimer (and, consequently, the form of protomers) is conserved among all three apo structures, there are local regions that adopt slightly diverse conformations. Most differences are due to different crystal packing that for spacegroups *C* 2 and *P* 2_1_ results to be quite tight, with V_M_ in the range of 2.0 Å^3^/Da (solvent content ~40%), while for *P* 2_1_2_1_2_1_ V_M_ is 2.6 Å^3^/Da, indicating a solvent content of ~50% (Appendix A). The loops between residues 44–62, 167–171 and 185–195, which are located at the entrance of the active-site region, show higher flexibility than the rest of the residues (Appendix A), as already reported by us and others [29,30,31].

We next obtained well-diffracting crystals of Mpro bound to MG-132 by exploring different co-crystallization conditions that resulted in solving structures from three different spacegroups: *C* 2, *P* 2_1_2_1_2_1_ and the recently reported *P* 1 [32]. Moreover, we obtained crystals in the presence and absence of DTT. Appendix A summarizes data collection and refinement statistics for all the apo and MG-132 bound structures here presented. In all the structures the ligand is covalently bound to the catalytic Cys145, independently from the crystallization conditions, crystal packing and from the presence or absence of DTT (see Appendix A with the electron density of each structure). As further proof of the stabilizing binding of MG-132 to the Mpro construct used in the crystallization trials, we performed TSA analysis at different concentrations of MG-132. The thermal stability of the dimer increases by augmenting the inhibitor:enzyme ratio. At one molar excess, the ΔTm is only 0.5 °C and rises to 3.5 °C when the inhibitor is added at 50 molar excess. The increase has been normalized with the amount of DMSO present in the assay (Appendix A).

From the structural comparison between bound and unbound models, we noted that in spacegroup *C* 2 the binding site slightly opens to accommodate the MG-132 moiety (with a cell volume increase of +3% upon inhibitor binding). As a result, the cell parameters change with respect to the apo structure determined in the same spacegroup (Appendix A). As shown in Figure 1a, regions corresponding to residues 45–59, 164–169 and 185–195 move. In spacegroup *P* 2_1_2_1_2_1_, differently from what happens in spacegroup *C* 2, the binding site is already open enough to accommodate the inhibitor (Figure 1b,c—with a cell volume change <1%). As previously suggested by Kneller et al. in their room-temperature X-ray crystallography studies [31], in structures with bound inhibitors, the active site cavity shows high plasticity and often adopts more open conformations in comparison to the reported apo structures (all in spacegroups *C* 2 and *P* 2_1_). Notably, the apo structure we solved in *P* 2_1_2_1_2_1_ results to be rather open and to be even more similar to the inhibitor bound structures than the reported RT structure (PDB ID: 6WQF) (Appendix A). We, therefore, suggest that our new apo structure in spacegroup *P* 2_1_2_1_2_1_ (PDB ID: 7BB2) could be useful for future structure-based drug design and in silico docking studies.

The inhibitor bound structures obtained in spacegroup *P* 2_1_2_1_2_1_ in the presence and absence of DTT (PDB ID: 7BE7 and 7BGP, respectively) are almost identical having an overall RMSD of 0.084 Å. The same result is obtained with spacegroup *P* 1 (PDB ID: 7NG3 and 7NG6) with an RMSD of 0.124 Å, confirming the biochemical data showing that the inhibitory potency of MG-132 on Mpro is independent of DTT presence as shown in this paper and a previous one [10].

In general, MG-132 is almost identically bound in all the different spacegroups, with minor differences that can be attributed to the presence of symmetry contacts in spacegroups *C* 2 and *P* 1, while in spacegroup *P* 2_1_2_1_2_1_ the MG-132 moiety is not involved in symmetry contacts (see Appendix A). For this reason, coupled with the higher resolution (1.68 Å) of the crystals obtained in this spacegroup compared to the other two (1.94 Å in *C* 2, 1.8 Å-1.87 Å in *P* 1), we hereafter describe only the *P* 2_1_2_1_2_1_ structure obtained in presence of DTT (PDB ID: 7BE7).

The 2Fo-Fc map unambiguously shows the binding mode of MG-132 (Figure 2b) and is well defined in both protomers. The catalytic Cys145 attacks the aldehyde group of MG-132, forming a covalent hemithioacetal bound. A new chiral center is formed, and the reaction results to be stereoselective, leading to the (S) configuration which is the most typical conformation for Mpro aldehyde Cys-trap inhibitors [33,34,35]. MG-132 extends along the S1–S4 binding subsites of each protomer (Appendix A), interacting with residues through hydrogen bonds and hydrophobic interactions, in addition to the covalent bond with Cys145. (Figure 2c). The newly formed thiohemiacetal occupies the oxyanion hole formed by the backbone amide groups of Gly143, Ser144, and Cys145, where it forms hydrogen bonds with the amide groups of Gly143 and Cys145. The three Leucine residues of MG-132 respectively occupy the S1 (Leucine P1), S2 (Leucine P2) and S3 (Leucine P3) cavities, while the Z-group nicely occupies the S4 pocket. Hence, MG-132 interacts via hydrophobic interactions with residues Hys41, Glu166, Pro168 and Gln189. It also forms hydrogen bonds with the amide group of Glu166 as well as with the carbonyl group of residues Hys164 and Glu166, and with the amide side chain of Gln189.

### 2.2. Mpro Inhibition Mechanism by MG-132

To get insights into the mode of MG-132 related inhibition of Mpro we used our biochemical assay (performed independently by the groups at Fraunhofer Institute and at University of Cagliari) and to confirm the DTT independence and determine key kinetic and inhibitory parameters in the presence and absence of MG-132.

At first, we definitely assessed if a reducing environment could interfere with MG-132 activity, we determined IC_50_ in the same buffer reaction, in the presence/absence of DTT. Appendix A shows the dose dependence of the MG-132 compound in both conditions. The compound has a comparable inhibition profile and the IC_50_ value in the absence of DTT is similar to that calculated in the presence of DTT.

Determination of the Km and Vmax parameters show that without preincubation with MG-132 the Km increases by increasing inhibitor concentration (Figure 3A), reflecting a competitive situation. At the same time, we observe a drop in Vmax that would be typical for non-competitive inhibition. Knowing from the crystal structures that MG-132 is undoubtedly a covalent inhibitor and binds the substrate active site, the data hint towards an irreversible inhibition or a long dissociation constant koff. Measuring the same parameters generated after 1 h preincubation with MG-132, we observe that Vmax decreases significantly compared to Vmax without the inhibitor, but the Km remains almost unchanged. This supports the theory of long koff/irreversible binding mechanism (Figure 3B).

In the effort to further clarify the mechanism of action of MG-132 on Mpro, we measured the IC_50_ at different preincubation times (Figure 4A). In fact, if a compound reversibly binds an enzyme, it presents IC_50_ values that differ with a variation in the preincubation time [36]. Results showed that preincubation of MG-132 with MPro for 30, 45, and 120 min leads to an increase in IC_50_ values that are 3.8, 5.0 and 8.8 μM, respectively (Figure 4B). Noteworthy, *t*-test analysis showed a significant statistical *p*-value between 30 and 120 min of preincubation times (Figure 4C). To confirm this result, we performed a control experiment using GC376, which is reported to be a covalent but reversible binding inhibitor of Mpro [36]. Additionally, in this case, *t*-test analysis showed a significant statistical *p*-value in the same preincubation times observed for MG-132 (Figure 4F). The data reported in Figure 4D,E demonstrate that GC376 has a similar inhibition profile, with increased IC_50_ values, as determined for MG-132. These results are compatible with reversible inhibition of the SARS-2 MPro by MG-132.

To get a clearer separation between irreversible and reversible binding we further analyzed the impact on the IC_50_ value upon different incubation times of MG-132 with Mpro. We compared data generated with and without preincubation of MG-132 with the enzyme analyzing the curves generated within the first 20 min of reaction after substrate addition to avoid any effect of substrate depletion. According to obtained curves, measured at different time points of incubation with the substrate, IC_50_ values increase, showing loss of potency (Figure 5). In agreement with Krippendorf et al. [37] and with Yan et al. [38], irreversible inhibitors should show a decrease of IC_50_ representing a gain in the potency of the inhibitor.

Taken together, all our data obtained from performing different activity assays of Mpro in presence of MG-132, hint at a covalent reversible binding mode of MG-132 in the active side of Mpro with a long koff.

### 2.3. MG-132 and Cathepsin-L Inhibition

Knowing the role of MG-132 in inhibition of the proteasome and representatives of Cathepsin and Calpain proteases, we investigated its role in the inhibition of Cathepsin-L, one of the key enzymes needed for the entry of SARS-CoV-2. Previously, we reported the anti-cytopathic activity of MG-132 in SARS-CoV-2 infected Vero-E cells [see CHEMBL Entry https://www.ebi.ac.uk/chembl/document_report_card/CHEMBL4495565/, accessed on 28 October 2021]. The results were confirmed by different groups reporting antiviral activity also in other cell lines like Huh-7 and Calu-3 all with sub-micromolar EC50 [39]. If we compare the antiviral activity in cells to the high IC_50_ measured against Mpro it is reasonable to assume that the effect on Mpro is probably too low to justify submicromolar cell activity, and thus it is supported by inhibition of host proteases. We have measured no activity towards the second SARS-CoV-2 protease PLpro (paper in preparation).

Looking at dose-dependent MG-132 inhibition of Cathepsin-L using the BPSBioscience inhibitor screening kit, we saw a strong effect with 0.15 nM IC_50_ which corresponded to half of the used enzyme concentration in this biochemical assay (Figure 6). Previously, the inhibition of rat liver isolated Cathepsin-L was reported to be in the high nanomolar range (EC_50_ 163 nM) [40]. E64 was used as a positive control in the assay confirming literature values for Cathepsin-L inhibition with an IC_50_ of 2.7 nM (literature value IC_50_ = 2.5 nM; https://www.apexbt.com/e-64.html, accessed on 28 October 2021).

### 2.4. Docking of MG-132 in Cathepsin-L Binding Site

To investigate the interactions between MG-132 and Cathepsin-L consecutive molecular docking experiments were performed from induced-fit docking up to covalent docking (see methods). At the time of paper writing, 37 structures of Cathepsin-L were deposited in the Protein Data Bank (Appendix A), 25 of which are in complex with various ligands. Cathepsin-L is also structurally similar to the *Carica Papaya* papain protease (Figure 7), co-crystallized with MG-132, specifically in the region of the binding site, despite the overall sequence identity being relatively low (~40% and 57% in the binding site—see methods). Root mean square deviations (RMSD) of Cathepsin-L structures with respect to papain (PDB ID 1BP4) [41] vary between 0.623 Å and 0.782 Å.

Among these, we selected the one whose complexed ligand resembled more MG-132 according to the Tanimoto score of the ligand present [42]. This is 3OF9 in complex with Z-Phe-Tyr(t-Bu)-diazomethylketone [43]. After the first round of induced fit docking, two main orientations of MG-132 in the binding pocket were observed: orientation A where the benzyloxycarbonyl of MG-132 interacts with residues Leu70, Met71, Ala136, and Ala215 via hydrophobic contacts and orientation B where the aromatic moiety of the ligand interacts with Trp190, Glu193, and Trp194. The latter one is the same orientation that MG-132 has in papain (Figure 7b).

The formation of a covalent bond between MG-132 was further investigated by performing covalent docking (see methods for details). Based on the structure of the papain enzyme, we impose the covalent bond to happen between Cys26 of the protein and the aldehyde carbon atom of the ligand. The best-score (see method for details) structure with the ligand in the B orientation was selected (Figure 7b). The selected pose is stabilized by a pi-stacking interaction with Trp190 and by numerous hydrogen bonds with Gln20, Gly24, Cys26, Trp27 and Asp163, while the leucine delta carbon atoms are exposed to the solvent. This pose shows the ligand can smoothly fit into the binding site, and it is in line with the experimental nanomolar functional IC_50_ measured in our assay.

## 3. Discussion

Our work provides robust structural and biochemical data to elucidate the inhibition mechanism of MG-132 against SARS-CoV-2 Mpro. We solved seven crystal structures of Mpro in apo and inhibitor bound forms obtained from different conditions and in different spacegroups. By comparing all structures, we demonstrate that MG-132, being an aldehyde warhead peptidomimetic compound, has a well-defined binding mode that does not significantly perturb the overall fold of the dimer. The covalent stereoselective (S) hemithioacetal bond is nicely defined in the electron density maps of all our solved structures. From an extensive biochemical analysis this bond results to be reversible, as expected, but Km and Vmax measured at different incubation times suggest a slow koff, indicative of a long residency time. Our observations are in line with results from other groups [32,34,35].

In this study we also observe that the loops situated at the entrance of the active site require local movements, to allow the ligand to fit into the binding pocket and to react with the catalytic cysteine. The little differences we detect among the crystal forms confirm that Mpro shows local plasticity that was previously detected by molecular dynamics [29]. Here we present an apo structure of Mpro obtained in space group *P* 2_1_2_1_2_1_, which is more similar to the bound structures with respect to the other apo structures deposited in PDB that have spacegroups *C* 2 and *P* 2_1_. We, therefore, suggest that this structure could prove useful for future drug design and in silico efforts.

Last April, Pfizer announced that its oral Mpro inhibitor PF-07321332 entered Phase I clinical trials [44]. The molecule resembles DPP4 inhibitors with structural similarities with MG-132 [45] as it bears a nitrile group as a warhead for catalytic Cys instead of a carbonyl group and has optimized lipophilic P2–P3 moieties. A trifluoroacetyl group substitutes the carbobenzyloxy N-Terminal of MG-132 to enhance metabolic stability and oral bioavailability. No data has been shared about PF-07321332 selectivity and potency, yet. Nevertheless, this is a nice example of an optimized peptidomimetic as a drug candidate for rapid development against pandemic risks (a nice review has been recently published on all peptidomimetics so far investigated [46]).

In general, cysteine proteases have been progressively recognized as validated targets for the treatment of several human diseases [47]. The possibility either to combine Cathepsin-L and Mpro inhibitors or to identify, like in this case, dual inhibitors for both enzymes, even if not novel [48], has received attention from our group.

The most widely explored cysteine protease inhibitors use an electrophile to covalently modify the active cysteine and a recognition motif for binding to the active site; as is the case for MG-132. Selectivity is instead driven by groups facing P1 or P2 positions. In Mpro, the glutamine residue or γ-lactam residue are preferred P1 for its S1 pocket, while glycine is the most favored residue at the P1′ position. At the P2 position, Mpro favors leucine but it can accommodate other hydrophobic residues as well [28,49]. On the other side, for Cathepsin-L, specificity is predominantly guided by P2, which has a strong preference for aromatic and, to a slightly lesser extent, aliphatic residues [50,51]. While in P1, Cathepsin-L displays mixed selectivity for glutamine and glycine [50,51] that, however, is common to other Cathepsin proteins, like Cathepsin-B, as well as to Mpro. 

Therefore, in order to regulate the selectivity of MG-132 for Cathepsin-L and/or Mpro, the Leu sidechain facing P2 would be a key position to optimize. Our crystallographic and docking models can be instrumental to probe the S2 subsites of both targets.

An additional site to optimize the selectivity of dual Mpro/Cathepsin-L inhibitors is the cysteine reactive warhead as recently discussed by Ma et al. [52]. Substitutions of the aldehyde group with other reversible groups, such as a-keto derivatives or nitriles can be considered.

The main concern against a strategy to develop dual inhibitors is not in the potentially toxic effects of inhibitors in important host enzymes like Cathepsin-L, as tens of FDA-approved drugs possessing Cathepsin-L inhibiting activity have been on the market [48]. Nevertheless, the right in vivo pharmacokinetic profile is a much more challenging target for combination therapy than with dual agent monotherapy.

On the other side, lack of antiviral cytopathic effects, at least in some cells, of such dual inhibitors should alert medicinal chemists to the need to optimize such molecules preventing fast cellular degradation and raising their permeation at needed concentration levels in target cells (e.g., lung cells) at precise timing after infection or before it [53]. Whether preventive or curative, systemic or local delivery via nasal spray form of administration will be more effective is yet to be defined with further studies in animals.

## 4. Materials and Methods

### 4.1. Protein Expression and Purification

The Mpro of SARS-CoV-2 plasmid was kindly provided by the research group of Prof. Rolf Hilgenfeld from Institute of Biochemistry, Center for Structural and Cell Biology in Medicine, University of Lübeck (Germany) (ORF1ab polyprotein residues 3264–3569, GenBank code:MN908947.3) protein was expressed in *E. coli* and purified at homogeneity following the protocol reported in Zhang et al., 2020 [28]. Notably, the expression was done in E.coli BL21 (DE3) cultivated in YT medium supplemented with Ampicillin, inducing at OD600 of 0.8 with 0.5 mM IPTG at 37 °C for 5 h. Harvested cells were resuspended in lysis buffer (20 mM Tris pH 7.8, 150 mM NaCl) added with protease inhibitors (1 mM PMSF, 1 μg/mL Pepstatin A, 1 μg/mL Aprotinin, 4 μg/mL Leupeptin) and ruptured by high pressure homogenizer at 1000–1500 bar. The total cell extract was added by Nuclease (Pierce, Thermo Fischer Scientific, Waltham, MA, USA), clarified by centrifugation and loaded on a HisTrap FF Crude column (GE Healthcare, Chicago, IL, USA). The protein eluted fractions were pooled and dialyzed overnight against 20 mM Tris pH 7.8, 150 mM NaCl, 1 mM DTT buffer in presence of Prescission Protease. The dialyzed fraction was purified by negative affinity on GST-Protino resin (Macherey-Nagel, Dueren, Germany) and Ni-NTA resin (Qiagen, Hilden, Germany), subsequently. The obtained fraction was buffer exchanged and purified on 5 ml HiTrap Q HP. Pure protein samples were buffer exchanged in 20 mM Tris-HCl, 150 mM NaCl, 1 mM EDTA, pH 7.8 with or without 1 mM DTT, according to the final application. Sample aliquots were flash frozen in LN_2_ at a concentration of 10–20 mg/mL and stored at −80 °C.

### 4.2. Mpro Enzymatic Activity

The determination of Mpro key enzymatic parameters in the presence and absence of MG-132 and the validation of the inhibition mode of MG-132 against Mpro was measured in a Förster resonance energy transfer (FRET) assay as described in Kuzikov et al. [10]. Briefly, MG-132 (stock of 10 mM in 100% *v*/*v* DMSO) was transferred to black 384-well assay micro-plates (Corning, #3820, Corning, NY, USA) by acoustic dispensing (Echo, Labcyte, Beckman Coulter, Indianapolis, IN, USA). Dependent on the experiment Mpro was pre-incubated with MG-132 for 1 h at 37 °C or the FRET-substrate DABCYL-KTSAVLQ↓SGFRKM-EDANS (Bachem #4045664, Bubendorf, Switzerland) was directly added to the enzyme/compound mix. After 15 min of incubation at RT, generation of the EDANS-cleavage product was detected at Ex/Em = 340/460 nm (EnVision, PerkinElmer, Waltham, MA, USA). Final assay concentrations in 10 uL assay volume were: 60 nM Mpro, 15 μM FRET-substrate in assay buffer containing 20 mM Tris (pH 7.3), 100 mM NaCl, and 1 mM EDTA. Zinc pyrithione (MedChemExpress, #HY-B0572, Monmouth Junction, NJ, USA) 10 mM in 100% DMSO) was used as a positive control for 3CLpro inhibition. DMSO was used as compound solvent control (0% inhibition).

The determination of time-dependent inhibition of the MG-132 compound was evaluated in a biochemical assay as described in Kuzikov et al. [10]. Briefly, the enzymatic activity was established in a 384-plate format and measured in a buffer containing 20 mM Tris (pH 7.3), 100 mM NaCl and 1 mM EDTA; 100 nM of the enzyme was diluted in the presence/absence of DTT and different concentrations of compounds were incubated for 30/45/120 min at 37 °C and 15 μM of FRET substrate DABCYL-KTSAVLQ↓SGFRKM-EDANS (Bachem, Bubendorf, Switzerland) was added in each well, and after 15 min of incubation at 25 °C the fluorescent product was monitored (Ex/Em = 340/460 nm) (Nivo, PerkinElmer). GC376 was used as a positive control [54].

### 4.3. Cathepsin-L Inhibition Assay

The effect of MG-132 on Cathepsin-L activity was measured using the Cathepsin-L Inhibitor Screening Kit (Fluorometric) (BPS Bioscience #79591, San Diego, CA, USA). The assay is based on the Cathepsin-L mediated cleavage of a synthetic fluorogenic substrate. Generated AFC is measured on a fluorescence microplate reader. The assay was performed according to the manufacturer protocol but adapted to 384-well format with a final volume of 10 μL. Briefly, compounds (stock in 100% *v*/*v* DMSO) are transferred in black 384-well microplates by acoustic dispensing (Echo, Labcyte, Beckman Coulter); 5 µL of diluted Cathepsin-L is added and incubated for 30 min at RT. The enzymatic reaction is initiated by the addition of 5 µL of the substrate. The fluorescence signal is detected in a kinetic mode for 15 min at 37 °C using Ex/Em = 360/460 nM (Envision, PerkinElmer). Final assay concentrations were: Cathepsin-L 0.01 ng/μL, substrate 5 μM. E-64 provided in the kit was used as a positive control for Cathepsin-L inhibition and assay validation. DMSO was used as compound solvent control (0% inhibition).

### 4.4. Thermostability Assay

A reaction mixture containing the protein at 1 μM final concentration and increasing molar excesses of MG-132 (1×, 3×, 10×, 20×, and 50×) was tested. In each well: 20 μL of buffer 5×, 2 μL of protein stock, the volume of MG-132 stock needed for the fixed molar excess, and Milli-Q H_2_O to reach 99 μL was added. The protein was left in incubation with the compound for 30 min at room temperature. The plate was put on ice, and 1 μL of 500 × SYPRO orange (Thermo Fischer Scientific) was added in each well. The plate was sealed and centrifuged at 200× *g* and 4 °C for 1 min. The melting curves were measured in a RT-PCR (Eppendorf, Milan, Italy). After a stationary phase of 2 min at 25 °C, the temperature was increased by 0.5 °C every 30 s until the maximum of 95 °C was reached. Fluorescence has been measured at each step. Each condition has been reproduced in tripled. The Tm control with the protein and the equivalent volume of DMSO only was assessed as well.

### 4.5. Crystallization

All crystals were grown from SARS-CoV-2 Mpro at a concentration of 5 mg/mL in 20 mM Tris-HCl, 150 mM NaCl, 1 mM EDTA, pH 7.8 with or without 1 mM DTT. Apo form crystals in spacegroup *P* 2_1_ and *C* 2 were obtained at RT from several conditions of the PACT Premier™ screening (Molecular Dimensions, Portobello, Shieffield, UK). The best apo crystals in spacegroup *C* 2 and *P* 2_1_ were obtained in 0.1 M Na Acetate, 20% PEG 3350. Apo crystals in spacegroup *P* 2_1_2_1_2_1_ were obtained in several conditions of the Morpheus^®^ kit. The best diffracting one grew in: 0.1 M D-Glucose, 0.1 M D-Mannose, 0.1 M D-Galactose, 0.1 M L-Fucose, 0.1 M D-Xylose, 0.1 M N-Acetyl-D-Glucosamine, 0.1 M Imidazole/MES monohydrate (acid) pH 6.5, 20% *v*/*v* Ethylene glycol, 10% *w*/*v* PEG 8000.

Crystals of Mpro in complex with MG-132 were obtained at RT with the vapor diffusion technique, in sitting drops, using seeding in different conditions of the Morpheus^®^ kit and the ProPlex^®^ kit (Molecular Dimensions). Mpro and MG-132 were incubated for 1 h at room temperature prior to the set-up of the crystallization experiments, which were performed using a Mosquito robot (STPlabtech Ltd., Melbourn Hertfordshire, UK) mixing 50 nL of seeding solution with 220 nL of protein:MG-132 solution and 230 nL of the precipitant solutions. Crystals appeared in a few hours and were frozen after 1–2 days with LN_2_. The best diffracting crystals in spacegroup *P* 2_1_2_1_2_1_ both in presence and absence of DTT were obtained in condition E9 of the Morpheus screening: 0.12 M Ethylene glycols (Diethylene glycol; Triethylene glycol; Tetraethylene glycol; Pentaethylene glycol) 0.1 M Tris/BICINE pH 8.5, 20% *v*/*v* PEG 500 MME; 10% *w*/*v* PEG 20000. The best diffracting crystals in spacegroup *P* 1 in presence of DTT, and in spacegroup *C* 2 grew in: 0.05 M Magnesium chloride hexahydrate, 0.1 M MES pH 6.5, 5% *w*/*v* PEG 4000, 10% *v*/*v* 2-Propanol, while in absence of DTT they grew in: 0.1 M Magnesium acetate tetrahydrate, 0.1 M MES pH 6.5, 10% *w*/*v* PEG 10.000.

### 4.6. Data Collection, Data Reduction, Structure Determination, Refinement and Final Model Analysis

Diffraction data were collected at 100 K, at the XRD2 beamline of the Elettra synchrotron (Trieste, Italy) [55] using wavelengths ranging from 0.9716 to 1.000 Å. The collected datasets were processed with XDS [56] and Aimless [57] from the CCP4 suite [58]. The structure was solved by molecular replacement with Phaser [59] using as a search model 7BB2 (PDB ID). The initial models were refined alternating cycles of manual model building in COOT [60,61] and automatic refinement using Phenix [62] (version 1.19.2-4085) (http://www.phenix-online.org/, accessed on 28 October 2021). Data collection and refinement statistics are reported in Appendix A.

Figures were prepared using Pymol [The PyMOL Molecular Graphics System, Version 2.1 Schrödinger, LLC., New York, NY, USA]. RSMD between Cα of each residue was calculated using Superpose program from the CCP4 suite [58].

### 4.7. Molecular Docking

The protein (PDB ID: 3OF9) was preprocessed with the Protein Preparation Wizard from the Schrödinger Suite version 2020-4 [Schrödinger Release 2020-4: Schrödinger, LLC, New York, NY, USA] with the default parameters. The protonation state of the receptor was generated using Epik [63,64] for pH = 7 ± 2. All water molecules and co-crystallized molecules were removed. Following hydrogen bonds optimization, energy minimization was performed using the OPLS3 force field [65,66]. The molecule MG-132 was then retrieved from PubChem (compound 462382) in SDF format and prepared with the Ligprep tool [Schrödinger Release 2020-4: LigPrep, Schrödinger, LLC] (retaining the 3D chirality and generating states with Epik for pH = 7 ± 2) to be used in standard, induced-fit and covalent docking procedures (Glide version 84013, Schrodinger Python version 66013, Prime version 5.7, Covalent Docking v1.3) [67,68]. The induced-fit docking procedure, differently from a standard docking procedure, takes into account the flexibility of the residues in the binding site. Additionally, the covalent docking procedure is able to mimic the formation of a covalent bond between ligand and receptor. Covalent complexes are minimized using the Prime VSGB2.0 energy model to score the top covalent complexes. The grids for the docking were prepared using the default parameters, with the internal grid box centered on the centroid of the catalytic cysteine. The external grid box was defined by checking the option “Dock ligands similar in size” (~32× ~32× ~32 Å). For the induced-fit docking, the sidechains were trimmed automatically based on their B-factor. For the covalent docking, the aldehyde carbon atom was selected as the only reactive atom using SMARTS selection (“C(=O)H”). A maximum of 20 poses was produced for each run.

### 4.8. Sequence Alignment

The sequences of the structures 3OF9 [43] and 1BP4 [41] were obtained in FASTA format from the RCSB Protein Data Bank and aligned using the compositional score matrix adjustment method in Blastp [69,70]. Among the 84 residues in a 10 Å range from CYS26, 48 were identical between the two sequences.

### 4.9. Docking Validation 

The Glide XP docking procedure was validated in its ability to replicate the pose of a covalent inhibitor by re-docking the ligand AZ12878478 (RCSB ligand ID: NOW) into the binding site of Cathepsin-L (PDB id:3HHA) [71]. The RMSD between the predicted and experimental pose was 1.44 Å and calculated in Maestro Version 12.6.144 [Schrödinger Release 2020-4: Maestro, Schrödinger, LLC].

## Figures and Tables

**Figure 1 ijms-22-11779-f001:**
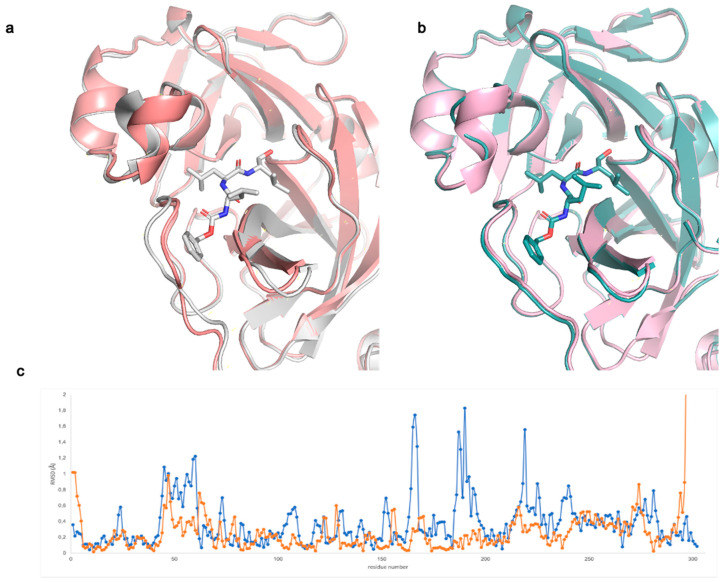
Comparison between Mpo apo and MG-132 complexed structures (**a**) in spacegroup *C* 2, grey (MG-132 bound, PDB ID: 7NF5) salmon (apo, PDB ID: 7ALH); (**b**) in spacegroup *P* 2_1_2_1_2_1_, teal (MG-132 bound, PDB ID: 7BE7) light pink (apo, PDB ID: 7BB2); (**c**) RSMD between Cα of each residue of the apo and MG-132 bound structures in spacegroup *C* 2 (blue) and spacegroup *P* 2_1_2_1_2_1_ (orange).

**Figure 2 ijms-22-11779-f002:**
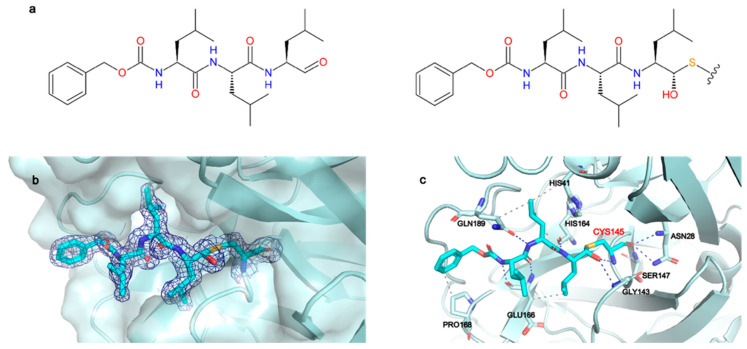
MG-132 is covalently bound to Cys145 in Mpro active site. (**a**) Left panel: chemical structure of MG-132, right panel: chemical structure of MG-132 covalently bound to Cys145. (**b**) X-ray crystal structure of MG-132 covalently bound to Cys145 with 2Fo-Fc map contoured at 1 sigma. (**c**) Main interactions of MG-132 with active site residues. Hydrogen bonds (blue dashed lines), hydrophobic interactions (grey dashed lines).

**Figure 3 ijms-22-11779-f003:**
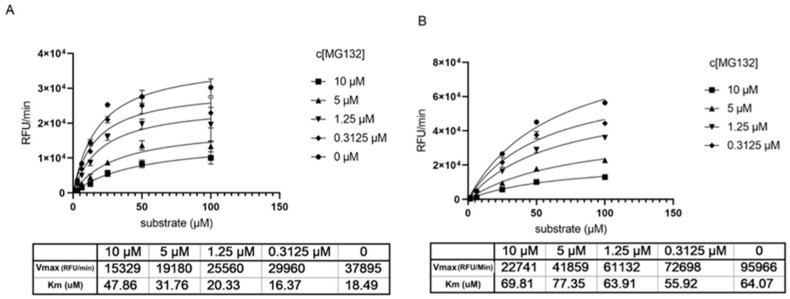
Determination of key kinetic parameters of Mpro in presence of MG-132. (**A**) without preincubation of Mpro with MG-132. (**B**) after 1 h preincubation of Mpro with MG-132.

**Figure 4 ijms-22-11779-f004:**
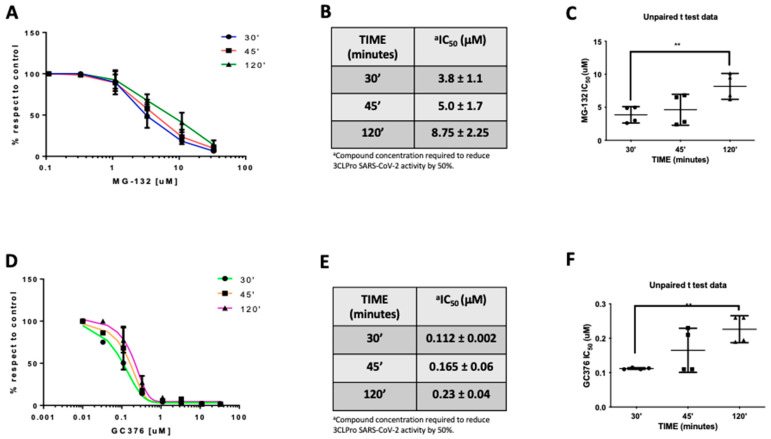
MG-132 and GC367 activities on SARS-CoV-2 MPro at different preincubation times. (**A**) MG-132 dose-dependent curves. (**B**) IC_50_ values (μM concentrations). (**C**) *t*-test of MG-132 IC_50_ values; (**D**) GC376 dose-dependent curves. (**E**) IC_50_ values (μM concentrations). (**F**) *t*-test of GC376 IC_50_ values.

**Figure 5 ijms-22-11779-f005:**
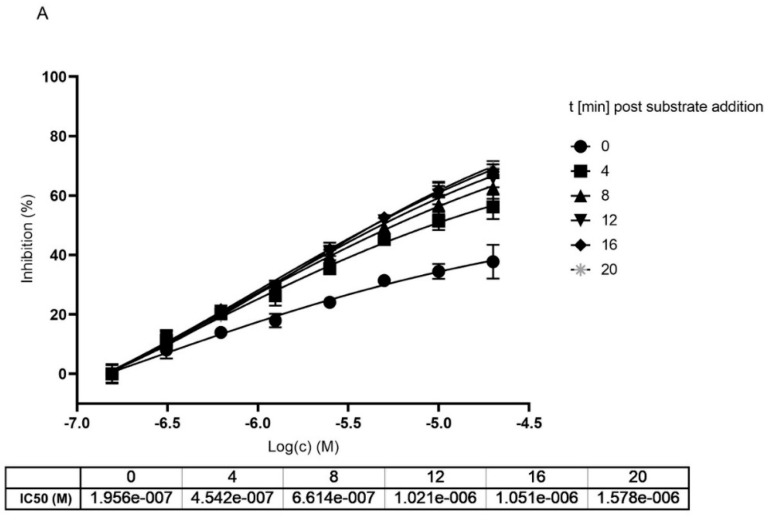
Dose and time dependent inhibition of Mpro using MG-132. (**A**) without preincubation of Mpro with MG-132. (**B**) after 1 h preincubation of Mpro with MG-132. Dose response curves were measured at different time points after substrate addition.

**Figure 6 ijms-22-11779-f006:**
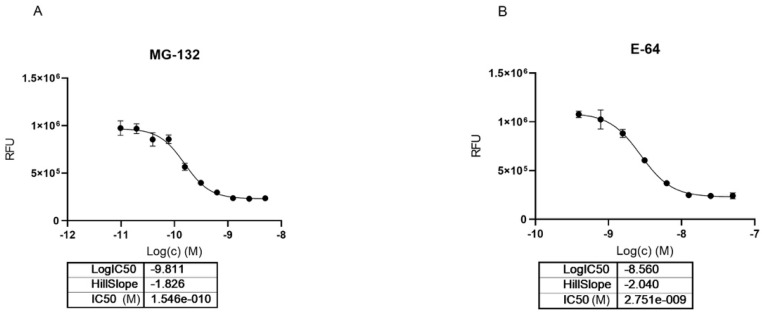
Dose dependent inhibition of Cathepsin-L using MG-132 (**A**) and E64 (**B**).

**Figure 7 ijms-22-11779-f007:**
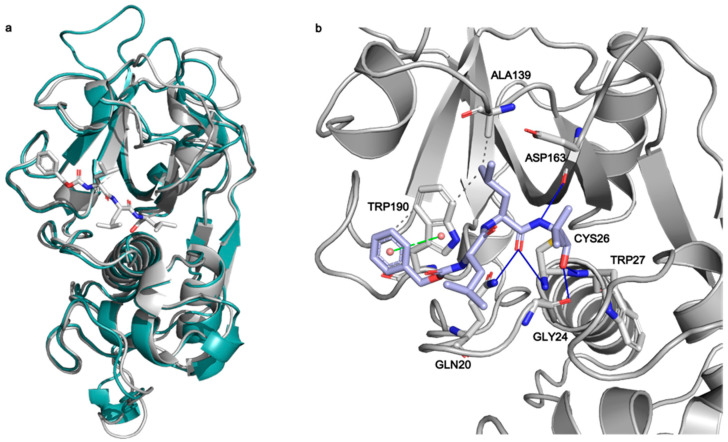
(**a**) Superposition of Cathepsin-L (3OF9, grey) and papain (1BP4, teal) structures. The MG-132 molecule co-crystallized in 1BP4 is depicted in white sticks. (**b**) Best scoring pose obtained from the covalent docking of MG-132 on Cathepsin-L. The main interactions of MG-132 are shown: hydrogen bonds (blue lines), hydrophobic interactions (grey dashed lines), pi-stacking interactions (green dashed lines).

## Data Availability

Data available in a publicly accessible repository. Coordinates and structure factors were deposited to the Protein Data Bank with accession numbers 7ALH (apo in spacegroup *C* 2), 7ALI (apo in spacegroup *P* 2_1_), 7BB2 (apo spacegroup in *P* 2_1_2_1_2_1_), 7BE7 (MG-132 covalently bound, in presence of DTT in spacegroup *P* 2_1_2_1_2_1_), 7BGP (MG-132 covalently bound, in absence of DTT, in spacegroup *P* 2_1_2_1_2_1_), 7NG3 (MG-132 covalently bound, in presence of DTT in spacegroup *P* 1), 7NG6 (MG-132 covalently bound, in absence of DTT, in spacegroup *P* 1), 7NF5 (MG-132 covalently bound in spacegroup *C* 2).

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
