# Peer review of "Structural and Biochemical Analysis of the Dual Inhibition of MG-132 against SARS-CoV-2 Main Protease (Mpro/3CLpro) and Human Cathepsin-L"

_ijms, 2021, doi:10.3390/ijms222111779_

Round 1

Reviewer 1 Report

This is a valid scientific manuscript which is focused on the hot topic i.e., inhibitors of  SARS-CoV-2 and it uses robust multidisciplinary approach.  It merits to be published.

Author Response

We are please for the positive assessment of the Reviewer 1 and we have no specific comments to add, beside being thankful for the support and appreciation of our work. 

Reviewer 2 Report

Elisa et al. characterized the dual inhibition of a synthetic peptide aldehydes MG-132 to its target SARS-CoV-2 Main Protease and host cathepsin-L using structural and biochemical analysis. Several crystal structures of apo-main protease and Mpro/MG-132 complex were solved. Interactions between cathepsin-L and MG-132 were analyzed by model docking. The experiments were well designed and produced convinced results. And most of the conclusions are supported by the results. While I have some suggestions here hope could help.

  1. Line 114

The statement “Functionally, Cathepsin-L cleaves the SARS-CoV-2 S protein into S1 and S2 subunits and proteolytically activates cell–cell fusion. Confirming the same mechanism previously reported on S protein from SARS-CoV.” may not be accurate.

Actually, unlike SARS-CoV, SARS-CoV-2 contains a furin cleavage site and cell entry of SARS-CoV-2 is preactivated by proprotein convertase furin. SARS-CoV-2 spike can be cleaved by furin into S1 and S2.

Cell entry mechanisms of SARS-CoV-2. PNAS, 2020, Jian Shang et al.)

And TMPRSS2 and lysosomal cathepsin can activate both SARS-CoV and SARS-CoV-2.

  1. Hoffmann et al., SARS-CoV-2 cell entry depends on ACE2 and TMPRSS2 and is blocked by a clinically proven protease inhibitor.

Cell 181, 271–280.e8 (2020).

  1. Ou et al., Characterization of spike glycoprotein of SARS-CoV-2 on virus entry and its immune cross-reactivity with SARS-CoV. Nat. Commun. 11, 1620 (2020).

And as shown in reference 22 of this manuscript, CTSL functionally cleaved the SARS-CoV-2 S protein into smaller fragments (not into S1 and S2), and CTSL efficiently enhanced SARS-CoV -2 infection.

  1. MG-132 seems to have a strong inhibiting effect against cathepsin-L, which probably means they also have a strong binding affinity. So did the authors try crystal structure of the cathepsin-L/MG-132 complex rather than using the docking model.

  1. Regarding the biomedical assays in the manuscript, most of the assays were about the inhibiting effects between MG-132 and Mpro. Only figure6 was related to cathepsin-L, this maybe a little weak. Could the authors explain? And the IC50 values are not labeled in figure6.

  1. Figure4, does the 30” actually means 30 min? And the IC50 does not have the unit. And the six panels may need rearrangement to be better presented.

  1. In figure3, Vmax and Km do not have unit. Figure 5, IC50 does not have the unit.

  1. This study suggested a scaffold for the development of double-inhibitors. So based on the two complex structures, could the authors discuss about the optimization of MG-132, what maybe favorable for binding to both Mpro and cathepsin-L and what maybe favorable for one target while unfavorable for another one?

Author Response

Dear Reviewer

we are thankful for your comprehensive comments and suggestions. Below you will find, in red, our point-by-point reply to each question explaining our motivations and the corresponding correction/integration in the main text. We believe your comments have helped to improve our work. We are confident to have fulfilled your requirements and expectations.   

Point 1: Line 114

The statement “Functionally, Cathepsin-L cleaves the SARS-CoV-2 S protein into S1 and S2 subunits and proteolytically activates cell–cell fusion. Confirming the same mechanism previously reported on S protein from SARS-CoV.” may not be accurate.

Actually, unlike SARS-CoV, SARS-CoV-2 contains a furin cleavage site and cell entry of SARS-CoV-2 is preactivated by proprotein convertase furin. SARS-CoV-2 spike can be cleaved by furin into S1 and S2. (Cell entry mechanisms of SARS-CoV-2. PNAS, 2020, Jian Shang et al.) and TMPRSS2 and lysosomal cathepsin can activate both SARS-CoV and SARS-CoV-2.

  1. Hoffmann et al., SARS-CoV-2 cell entry depends on ACE2 and TMPRSS2 and is blocked by a clinically proven protease inhibitor.

Cell 181, 271–280.e8 (2020).

  1. Ou et al., Characterization of spike glycoprotein of SARS-CoV-2 on virus entry and its immune cross-reactivity with SARS-CoV. Nat. Commun. 11, 1620 (2020).

 And as shown in reference 22 of this manuscript, CTSL functionally cleaved the SARS-CoV-2 S protein into smaller fragments (not into S1 and S2), and CTSL efficiently enhanced SARS-CoV -2 infection.

 Response 1:

We thank the referee for pointing out the inaccuracy of our statement. We modified it in order to summarize the published data on SARS-2 spike protein proteolysis by Cathepsin-L and we updated the literature accordingly with the referee suggestions (see line 130-134, and reference 23,24,26].  

Point 2:

MG-132 seems to have a strong inhibiting effect against cathepsin-L, which probably means they also have a strong binding affinity. So, did the authors try crystal structure of the cathepsin-L/MG-132 complex rather than using the docking model.

Response 2: We thank the referee for this question. Indeed, the possibility of solving the crystal structure of the Cathepsin-L/ MG-132 complex was considered, however it was not possible to obtain highly pure “crystal grade” Cathepsin-L protein to perform crystallographic experiments. Therefore, to get hints of the docking pose of MG-132 into Cathepsin-L we choose to exploit the already existing collaboration with experts in computational modeling. They used with high confidence the Cathepsin-L crystal structures available in PDB and compared with the homologous structure of a papain protease bound to MG-132, as is described in the paper. We believe that the docking model obtained offers good hints for the design of optimized MG-132 like inhibitors. Of course, for the future, we might consider to find a strategy (albeit including other experts) to obtain crystallizable protein in complex with new inhibitors.

Point 3:

Regarding the biomedical assays in the manuscript, most of the assays were about the inhibiting effects between MG-132 and Mpro. Only figure 6 was related to cathepsin-L, this maybe a little weak. Could the authors explain? And the IC50 values are not labeled in figure6.

 Response 3: In this work we focus on characterizing the binding mode and the inhibition mechanism of MG-132 towards Mpro. Therefore, we performed a number of independent crystallographic and biochemical experiments to show that MG-132 is indeed binding in a well-defined mode into the Mpro active site, and to experimentally validate that the binding is reversible, as expected from the literature. This was the main scope of our work.

Therefore, we determined the IC50 value in a biochemical assay using a commercial protein and assay kit, in order to confirm previously reported IC50 (see ref 41 in main text). Given the obtained results, our expectation was that the mode of action would be conserved and we preferred to explore, at speculative level, how MG-132 would bind into cathepsin-L active site, since it shares some similar features with the above reported inhibitors. Our in silico evaluation suggests that this is indeed the case and such possibility open up potential dual-inhibition strategy based on MG-132 scaffold, that we would like to discuss for a follow-up paper. The study and the characterization of cathepsin-L binding profiles and function with MG-132, is indeed not in the scope of the current manuscript.

Figure 6 has been modified as requested.

Point 4:

Figure4, does the 30” actually means 30 min? And the IC50 does not have the unit. And the six panels may need rearrangement to be better presented.

Response 4:

Thank you for noting the mistake, indeed the time is 30 min. Figure 4 has been modified correcting the error in the inset and adding the IC50 units. 

Point 5:

In figure3, Vmax and Km do not have unit. Figure 5, IC50 does not have the unit.

Response 5: Figures 3 and 5 have been modified as requested, adding the missing units.

Point 6:

This study suggested a scaffold for the development of double-inhibitors. So based on the two complex structures, could the authors discuss about the optimization of MG-132, what maybe favorable for binding to both Mpro and cathepsin-L and what maybe favorable for one target while unfavorable for another one?

Response 6: In order to highlight the opportunity to use our structures to discuss optimization strategies of MG-132, we have added a paragraph in the discussion (lines 398-413) undelaying the present understandings of the preferred recognition motif for each enzyme, also including four more papers in the literature (see ref 49-52).  Nevertheless, as said above, we didn’t enter in a detailed analysis of favorable/unfavorable decorations that could be tested for lead optimization, as this was not the main scope of this paper and we would leave this discussion for a future and more focused study.

Round 2

Reviewer 2 Report

The authors has addressed all my questions. But some minor mistakes should be revised, some time points are still shown as 30'' in the figures.